# Accelerated Training of Physics-Informed Neural Networks (PINNs) using Meshless Discretizations

**Ramansh Sharma**
Department of Computer Science and Engineering, SRM Institute of Science and Technology, India
`rs7146@srmist.edu.in`

**Varun Shankar**
School of Computing, University of Utah, UT, USA
`shankar@cs.utah.edu`

## Abstract

Physics-informed neural networks (PINNs) are neural networks trained by using physical laws in the form of partial differential equations (PDEs) as soft constraints. We present a new technique for the accelerated training of PINNs that combines modern scientific computing techniques with machine learning: discretely-trained PINNs (DT-PINNs). The repeated computation of the partial derivative terms in the PINN loss functions via automatic differentiation during training is known to be computationally expensive, especially for higher-order derivatives. DT-PINNs are trained by replacing these exact spatial derivatives with high-order accurate numerical discretizations computed using meshless radial basis function-finite differences (RBF-FD) and applied via sparse-matrix vector multiplication. While in principle any high-order discretization may be used, the use of RBF-FD allows for DT-PINNs to be trained even on point cloud samples placed on irregular domain geometries. Additionally, though traditional PINNs (vanilla-PINNs) are typically stored and trained in 32-bit floating-point (fp32) on the GPU, we show that for DT-PINNs, using fp64 on the GPU leads to significantly faster training times than fp32 vanilla-PINNs with comparable accuracy. We demonstrate the efficiency and accuracy of DT-PINNs via a series of experiments. First, we explore the effect of network depth on both numerical and automatic differentiation of a neural network with random weights and show that RBF-FD approximations of third-order accuracy and above are more efficient while being sufficiently accurate. We then compare the DT-PINNs to vanilla-PINNs on both linear and nonlinear Poisson equations and show that DT-PINNs achieve similar losses with 2-4x faster training times on a consumer GPU. Finally, we also demonstrate that similar results can be obtained for the PINN solution to the heat equation (a space-time problem) by discretizing the spatial derivatives using RBF-FD and using automatic differentiation for the temporal derivative. Our results show that fp64 DT-PINNs offer a superior cost-accuracy profile to fp32 vanilla-PINNs, opening the door to a new paradigm of leveraging scientific computing techniques to support machine learning.

## 1   Introduction

Partial differential equations (PDEs) provide a convenient framework to model a large number of phenomena across science and engineering. In real-world scenarios, PDEs are typically challenging or impossible to solve using analytical techniques, and must instead be approximately solved using a numerical method. A variety of numerical methods to solve these PDEs have been developed including but not limited to finite difference (FD) methods (19) (which work primarily on rectangular domains

36th Conference on Neural Information Processing Systems (NeurIPS 2022).

partitioned into Cartesian grids) and finite element (FE) methods (36) (which work on domains with curved boundaries but require partitioning the domain into multidimensional simplices). A modern class of numerical methods called *meshless* or *meshfree* methods generalizes finite difference methods in such a way as to remove the dependence on Cartesian grids, thereby allowing for the numerical solution of PDEs on point clouds. Of these, radial basis function-finite differences (RBF-FD) are among the most popular and widely-used (3; 5; 37; 8; 9; 1; 11; 12; 13; 10; 25; 26; 14; 33; 18), though a host of other such methods also exist. Much like FD or FE methods, these meshless methods can also approximate solutions to a desired *order* of accuracy.

More recently, PDE solvers based on machine learning (ML) have begun to gain in popularity due to the inherent ability of ML techniques such as neural networks (NNs) to recover highly complicated functions from data specified at arbitrary locations (15; 20). We focus on a popular class of ML-based meshless methods called physics-informed neural networks (PINNs) (27). PINNs can be used both to discover/infer PDEs that govern a given data set, and as direct PDE solvers. Our focus in this work is on the latter problem, though our techniques extend straightforwardly to inferring PDEs as well. PINNs are typically multilayer feedforward deep NNs (DNNs) that are trained using PDEs and boundary conditions as soft constraints, leveraging automatic differentiation (autograd) for computing derivatives appearing in the PDE terms. The original PINNs, often referred to as vanilla-PINNs, are challenging to train, at least partly because PDE-based constraints lead to complicated loss landscapes (17). These issues are somewhat ameliorated by using domain decomposition (X-PINNs) (16) or gradient-enhanced training (G-PINNs) (38). Other approaches for ameliorating these issues involve curriculum training or sequence-to-sequence learning (17). Many of these extensions can also help improve training and test accuracy. Much like other DNNs, PINNs are typically trained in 32-bit floating-point (*i.e.*, fp32 or single precision).

In this work, we introduce a new technique for accelerating the training of vanilla-PINNs. Our technique relies on two key features: (a) using RBF-FD to compute highly accurate (nevertheless approximate) **spatial** derivatives in place of autograd, and (b) training the DNN in fp64 rather than fp32. These new discretely-trained PINNs (DT-PINNs) can be trained significantly faster than fp32 vanilla-PINNs on consumer desktop GPUs with no loss in accuracy or change in DNN architecture. The use of RBF-FD allows DT-PINNs to retain the meshless nature of vanilla-PINNs, thereby allowing for the solution of PDEs on domains with curved boundaries. As RBF-FD uses sparse-matrix vector multiplication (SpMV) to approximate the derivatives, DT-PINNs are also parallelizable on modern GPU architectures. It is important to note that DT-PINNs use autograd for the actual optimization of the PINN weights; only PDE derivatives are discretized using RBF-FD.

The NN literature does contain efforts to replace automatic differentiation with numerical differentiation. For instance, recent work showed that FD approximations can be efficient for learning generative models via score matching (23). Another example is an NN architecture that involves learning FD-like filters for faster prediction of PDEs (35). In the PINN literature, fractional-PINNs (F-PINNs) use numerical differentiation as autograd cannot compute fractional derivatives (22). Nevertheless, to the best of our knowledge, ours is the first work on using meshless high-order accurate FD-like methods in conjunction with PINNs, allowing them to be trained **without any loss in accuracy** on domains with curved boundaries (just as autograd does). An alternative would involve eliminating autograd inefficiencies via Taylor-mode differentiation (4). However, we show that at least part of the speedups observed in DT-PINNs is because numerical differentiation results in training completing in fewer epochs than if autograd were to be used.

To alleviate concerns about replacing autograd with RBF-FD, we first compare fp64 RBF-FD approximation of different orders of accuracy against fp32 autograd for DNNs and show the cost benefits of using higher-order accurate RBF-FD. Then, to illustrate the features of DT-PINNs, we focus for brevity on two purely spatial PDEs (the nonlinear and linear Poisson equations) and one space-time PDE (the heat equation). We use these settings to compare DT-PINNs and vanilla-PINNs for relative errors, timings, and speedups on a simple desktop GPU. We demonstrate through our experiments that DT-PINNs offer a superior cost-accuracy profile over vanilla-PINNs.

The remainder of this paper is organized as follows. In Section 2, we review both vanilla-PINNs and RBF-FD. Next, in Section 3, we discuss how to train DT-PINNs to solve both the Poisson and heat equations. Then, in Section 4, we present experimental results comparing RBF-FD and autograd, and comparing DT-PINNs against vanilla-PINN on the Poisson and heat equations. We summarize

our results and discuss possible future work in Section 5. Finally, the appendix contains additional results, code snippets, and key implementation details.

**Notation**: We use $x$ to refer to spatial coordinates in $d$ dimensions. On the other hand, a bolded quantity such as $\mathbf{c}$ or $\boldsymbol{u}$ indicates a vector with more than $d$ elements (an array). Finally, the $\sim$ symbol on top of a quantity indicates that the quantity is an approximation.

**Broader Impacts**: To the best of the authors' knowledge, there are no negative societal impacts of our work including potential malicious or unintended uses, environmental impact, security, or privacy concerns.

## 2 Review

We now provide a brief mathematical review of both vanilla-PINNs and RBF-FD discretizations. Unless we note otherwise, all derivatives in this section are spatial or temporal. We focus on three prototypical PDEs: the nonlinear Poisson equation, the linear Poisson equation, and the heat equation.

### 2.1 Physics-informed neural networks

Let $\Omega \subset \mathbb{R}^d$ be a domain with boundary given by $\partial\Omega$; here, $d$ is the spatial dimension. We will focus on the solution of the nonlinear Poisson equation on $\Omega$ using PINNs. Let $x \in \mathbb{R}^d$, and let $u : \mathbb{R}^d \to \mathbb{R}$ be the solution to

$$\Delta u(x) = e^{u(x)} + f(x), \ x \in \Omega, \tag{1}$$

$$(\alpha n(x) \cdot \nabla + \beta) u(x) = g(x), \ x \in \partial\Omega, \tag{2}$$

where $\Delta$ is the Laplacian in $\mathbb{R}^d$, $\nabla$ is the $\mathbb{R}^d$ gradient, $n(x)$ is the unit outward normal vector on the boundary $\partial\Omega$, $f(x)$ and $g(x)$ are known functions, and $\alpha, \beta \in \mathbb{R}$ are known coefficients. If the $e^{u(x)}$ term is dropped from (1), we obtain the simpler *linear* Poisson equation:

$$\Delta u(x) = f(x), \ x \in \Omega. \tag{3}$$

The vanilla-PINN technique for solving either Poisson problem involves approximating the unknown solution $u(x)$ by a DNN $\tilde{u}(x, \mathbf{w})$ (where $\mathbf{w}$ is a vector of unknown NN weights), so that $\|\tilde{u}(x, \mathbf{w}) - u(x)\| \leq \epsilon$ for some norm $\|.\|$ and some tolerance $\epsilon$. In the absence of existing solution data, this is accomplished by enforcing (1) and (2) as soft constraints on $\tilde{u}(x)$ to find the weights $\mathbf{w}$ during training. Denote by $X = \{x_k\}_{k=1}^N$ the set of training points at which these constraints are enforced; in the context of PDEs, these are also called **collocation points**. For convenience, we divide $X$ into two sets: $N_i$ interior points in the set $X_i$ and $N_b$ boundary points in the set $X_b$; then, $X = X_i \cup X_b$, and $N = N_i + N_b$. Further, let $\mathcal{B} = \alpha n(x) \cdot \nabla + \beta$. The vanilla-PINN training loss $e(x, \mathbf{w})$ can then be written as:

$$e(x, \mathbf{w}) = \underbrace{\frac{1}{N_i} \sum_{j=1}^{N_i} \left( \Delta\tilde{u}(x, \mathbf{w})|_{x=x_j} - e^{\tilde{u}(x_j)} - f(x_j) \right)^2}_{\text{PDE loss in interior}} + \underbrace{\frac{1}{N_b} \sum_{i=1}^{N_b} \left( \mathcal{B}\tilde{u}(x, \mathbf{w})|_{x=x_i} - g(x_i) \right)^2}_{\text{Boundary condition loss on boundary}}, \tag{4}$$

where $\Delta$ and the $\nabla$ term in $\mathcal{B}$ are both applied through autograd. The $\texttt{tanh}$ activation function is typically used, *L-BFGS* is used as the optimizer for finding the weights $w$, and training is typically done in fp32 (17). For the linear Poisson equation, one simply omits the $e^{\tilde{u}}$ term from the loss above.

For time-dependent PDEs, the PINN becomes a function of space and time $\tilde{u}(x, t)$. We focus on the forced heat equation, given by

$$\frac{\partial u(x, t)}{\partial t} = \Delta u(x, t) + f(x, t), \ x \in \Omega, \tag{5}$$

$$\mathcal{B}u(x, t) = g(x, t), \ x \in \partial\Omega, \tag{6}$$

$$u(x, 0) = u_0(x), \tag{7}$$

where (7) is an initial condition and $u_0(x)$ is some known function. While the $\Delta$ term is handled via autograd, there are two options to handle temporal derivatives: in a continuous fashion or a time-discrete fashion. In the former, one samples the full space-time interval $\Omega \times [0, T]$ with collocation/training points, and then uses autograd to compute all spatial and temporal derivatives. The loss terms are also augmented with the initial condition (7), which is enforced on the full space-time solution. In the time-discrete approach, one typically discretizes the time derivative using an appropriate scheme (such as a Runge-Kutta method), and then proceeds in a step by step fashion. We focus on the continuous approach in this work.

## 2.2 Radial basis function-finite differences (RBF-FD)

We now briefly review RBF-FD methods. Given some function $f : \mathbb{R}^d \to \mathbb{R}$, the goal of any FD formula is to approximate the action of a linear operator $\mathcal{L}$ on that function (*i.e.*, to approximate $\mathcal{L}f$) at some location $x_1$. This is typically accomplished by using a weighted linear combination of $f$ at $x_1$ and its $n - 1$ nearest neighbors. Mathematically, this can be written as:

$$\mathcal{L}f(x)|_{x=x_1} \approx \sum_{k=1}^{n} c_k f(x_k), \tag{8}$$

where the real numbers $c_k$ are called FD weights, and the set of points $x_1, \ldots, x_n$ is called an FD stencil. In general, given a set of samples $X = \{x_j\}_{j=1}^N$, one can repeat the above procedure to find FD weights at every single point. These weights can be assembled into an $N \times N$ *differentiation matrix* $L$ so that $\mathcal{L}f(x)|_X \approx L f(x)|_X$. If $n << N$, $L$ will be a sparse matrix with at most $n$ non-zero elements per row. If $X$ lies on a Cartesian grid, the entries of $L$ (*i.e.*, the FD weights $c_k$) are known in advance. However, if $X$ is a more general point cloud, standard FD cannot be used to generate the entries of $L$ (see Mairhuber-Curtis theorem (7)). The RBF-FD method involves using an interpolatory combination of RBFs and polynomials instead. Without loss of generality, we describe the RBF-FD procedure for $x_1$ and its $n - 1$ nearest neighbors. Let $\phi(r) = r^m$, where $m$ is odd, be a *radial kernel* (a polyharmonic spline), and $q_j(x)$, $j = 1, \ldots, \binom{\ell+d}{d}$ be a basis for polynomials of total degree $\ell$ in $d$ dimensions; we use tensor-product Legendre polynomials. The RBF-FD weights for the operator $\mathcal{L}$ at the point $x_1$ are computed by solving the following dense (block) linear system on this stencil:

$$\begin{bmatrix} A & P \\ P^T & 0 \end{bmatrix} \begin{bmatrix} \mathbf{c} \\ \lambda \end{bmatrix} = \begin{bmatrix} \mathcal{L}a \\ \mathcal{L}q \end{bmatrix}, \tag{9}$$

where

$$A_{ij} = \phi(\|x_i - x_j\|), i, j = 1, \ldots, n, \quad P_{ij} = q_j(x_i), i = 1, \ldots, n, j = 1, \ldots, \binom{\ell+d}{d}, \tag{10}$$

$$\mathcal{L}a = \mathcal{L}\phi(\|x - x_j\|)|_{x=x_1}, \quad \mathcal{L}q = \mathcal{L}q_j(x)|_{x=x_1}, j = 1, \ldots, \binom{\ell+d}{d}, \tag{11}$$

where $\mathbf{c}$ is the (column) vector of $n$ RBF-FD weights. The vector $\lambda$ is a set of Lagrange multipliers enforcing the condition $P^T \mathbf{c} = \mathcal{L}q$, thereby ensuring that (a) the RBF-FD weights $\mathbf{c}$ can **exactly** differentiate all polynomials up to total degree $\ell$; and that (b) the error in the RBF-FD approximation to $\mathcal{L}$ when applied to all other functions is $O(h^{\ell+1-\theta})$, where $0 \leq h \leq 1$ is a measure of sample spacing in the stencil, and $\theta$ is the number of derivatives in the differential operator $\mathcal{L}$ (6). We set $\ell = p + \theta - 1$ based on the desired order of convergence $p$ so that the error is $O(h^p)$. We then set the stencil size to $n = 2\binom{\ell+d}{d} + 1$ as this ensures that (9) has a solution (2), and also set $m = \ell$ if $\ell$ is odd, and $m = \ell - 1$ if $\ell$ is even (29). $L$ becomes more dense for higher values of $p$ and dimension $d$, as $n = O(p^d)$. When this procedure is repeated for each point in the set $X$, the cost scales as $O(N)$ for fixed $n$, with large speedups possible by computing multiple sets of weights using each stencil (28; 29; 31; 34; 32). For domains with fixed boundaries, the RBF-FD weights can be precomputed and reused during simulation. However, domains with moving boundaries require recomputation of RBF-FD weights proximal to the boundary every time-step; fortunately, this can be done quite efficiently (32).

**Ghost points** When tackling boundary conditions involving derivatives (such as in (2)) using RBF-FD, it is common to include a set of $N_b$ *ghost points* outside the domain boundary $\partial\Omega$ into the set of samples to ensure that RBF-FD stencils at the boundary are less one-sided; this aids in numerical stability and accuracy. Ghost points allow us to also enforce the PDE at both the interior and boundary points. We therefore define and use the extended set $\tilde{X} = X_i \cup X_b \cup X_g$, where $X_g$ is the set of ghost points. For the remainder of this article, let the RBF-FD differentiation matrix for $\Delta$ be $L$ (dimensions $(N_i + N_b) \times (N_i + 2N_b)$), and for $\mathcal{B}$ be $B$ (dimensions $N_b \times (N_i + 2N_b)$).

## 3 Discretely-Trained PINNs (DT-PINNs)

Having described both vanilla-PINNs and RBF-FD, we are now ready to describe DT-PINNs. In short, DT-PINNs are PINNs that are trained using the sparse differentiation matrices $L$ and $B$ in place of the autograd operations used to compute the Laplacian and boundary operators in the loss function (4) (and its heat equation equivalent). All operations are carried out in fp64.

**Poisson Equation** Focusing first on the nonlinear Poisson equation (1), recall that $\tilde{u}(x, \mathbf{w})$ is the PINN approximation to the true solution $u(x)$. Let the evaluation of $\tilde{u}(x, \mathbf{w})$ on the set $\tilde{X}$ be $\tilde{\boldsymbol{u}}$, *i.e.*, $\tilde{\boldsymbol{u}}$ is obtained by evaluating $\tilde{u}(x, \mathbf{w})$ at interior, boundary, *and* ghost points. Further define the vector $\mathbf{e}$, which is the loss function evaluated at only the interior and boundary points, *i.e.*, $\mathbf{e} = e(x, \mathbf{w})|_X$. Then, the DT-PINN loss function can be written as:

$$\mathbf{e} = \underbrace{\frac{1}{N_i + N_b}\|L\tilde{\boldsymbol{u}} - \exp(\tilde{\boldsymbol{u}}) - \mathbf{f}\|_2^2}_{\text{PDE loss in interior and on boundary}} + \underbrace{\frac{1}{N_b}\|B\tilde{\boldsymbol{u}} - \mathbf{g}\|_2^2}_{\text{Boundary condition loss on boundary}}, \tag{12}$$

where $L$ and $B$ were defined previously, $\exp(\tilde{\boldsymbol{u}})$ is the element-wise exponential of the vector $\tilde{\boldsymbol{u}}$, and $\mathbf{f} = f(x)|_X$, and $\mathbf{g} = g(x)|_{X_b}$; here, $\mathbf{f}$ has dimension $(N_i + N_b) \times 1$, and $\mathbf{g}$ has dimension $N_b \times 1$. For efficiency, $L$ and $B$ can be precomputed using RBF-FD before the training process begins, and then simply multiplied with the vector $\tilde{\boldsymbol{u}}$ to obtain its numerical derivatives. The loss function (12) is then minimized over $\mathbf{w}$ as usual using autograd in conjunction with a suitable optimizer. For the linear Poisson equation (3), we simply drop the $\exp(\tilde{\boldsymbol{u}})$ term.

**Heat Equation** When using DT-PINNs for the heat equation, we demonstrate the flexibility of our method by using a mixed training technique where the time derivative is handled with autograd and the spatial derivatives are discretized with RBF-FD; this also allows us to bypass the Courant-Friedrichs-Lewy (CFL) constraint on the time-step. We carefully order the evaluations of the network so that $L$ and $B$ multiply the right quantities. Let $\tilde{u}(x, t, \mathbf{w})$ be the PINN, and recall that we have $N_t$ time steps over the interval $[0, T]$; in addition, we also have the initial condition at time $t = 0$, making for a total of $N_t + 1$ steps. Define $\tilde{\boldsymbol{u}}_k = \tilde{u}|_{x=\tilde{X}, t=k\triangle t}$, where $\triangle t$ is the timestep. This vector is the evaluation of $\tilde{u}$ on all spatial locations (including ghost nodes) for the $k$-th time slice. This definition in turn allows us to define two vectors, $\tilde{\boldsymbol{u}}_\Delta$ and $\tilde{\boldsymbol{u}}_\mathcal{B}$ as follows:

$$\tilde{\boldsymbol{u}}_\Delta = \begin{bmatrix} L\tilde{\boldsymbol{u}}_0 \\ L\tilde{\boldsymbol{u}}_1 \\ \vdots \\ L\tilde{\boldsymbol{u}}_{N_t} \end{bmatrix}, \quad \tilde{\boldsymbol{u}}_\mathcal{B} = \begin{bmatrix} B\tilde{\boldsymbol{u}}_0 \\ B\tilde{\boldsymbol{u}}_1 \\ \vdots \\ B\tilde{\boldsymbol{u}}_{N_t} \end{bmatrix}. \tag{13}$$

The vector $\tilde{\boldsymbol{u}}_\Delta$ has dimensions $(N_t + 1)(N_i + N_b) \times 1$, and $\tilde{\boldsymbol{u}}_\mathcal{B}$ has dimensions $N_t N_b \times 1$. Next, we define the data vectors $\mathbf{f}$ and $\mathbf{g}$ as follows:

$$\mathbf{f} = \begin{bmatrix} \mathbf{f}_0 \\ \mathbf{f}_1 \\ \vdots \\ \mathbf{f}_{N_t} \end{bmatrix}, \quad \mathbf{g} = \begin{bmatrix} \mathbf{g}_0 \\ \mathbf{g}_1 \\ \vdots \\ \mathbf{g}_{N_t} \end{bmatrix}, \tag{14}$$

where $\mathbf{f}_k = f(x, t)|_{x=X, t=k\triangle t}$, and $\mathbf{g}_k = g(x, t)|_{x=X_b, t=k\triangle t}$. Finally, we define two more vectors: $\boldsymbol{u}_0 = u_0(x)|_X$, the vector evaluating the initial condition on the set $X$ (interior and boundary points); and $\tilde{\boldsymbol{u}}_t$, the vector of evaluations of $\frac{\partial \tilde{u}}{\partial t}$ at spatial locations (interior and boundary) for each time slice:

$$\tilde{\boldsymbol{u}}_t = \begin{bmatrix} \left(\frac{\partial \tilde{\boldsymbol{u}}}{\partial t}\right)_0 \\ \left(\frac{\partial \tilde{\boldsymbol{u}}}{\partial t}\right)_1 \\ \vdots \\ \left(\frac{\partial \tilde{\boldsymbol{u}}}{\partial t}\right)_{N_t} \end{bmatrix}, \tag{15}$$

where $\left(\frac{\partial \tilde{\boldsymbol{u}}}{\partial t}\right)_k = \frac{\partial \tilde{u}}{\partial t}\big|_{x=X, t=k\triangle t}$. This vector is computed using autograd. With these different vectors defined, we can finally write the DT-PINN loss vector $\mathbf{e}$ for the heat equation as

$$\mathbf{e} = \underbrace{\frac{1}{N_i + N_b}\|\boldsymbol{u}_0 - \tilde{u}|_{x=X, t=0}\|_2^2}_{\text{Initial condition}} + \underbrace{\frac{1}{(N_t + 1)(N_i + N_b)}\|\tilde{\boldsymbol{u}}_t - \tilde{\boldsymbol{u}}_\Delta - \mathbf{f}\|_2^2}_{\text{PDE loss in interior and on boundary}} + \underbrace{\frac{1}{(N_t + 1)N_b}\|\tilde{\boldsymbol{u}}_\mathcal{B} - \mathbf{g}\|_2^2}_{\text{Boundary condition loss on boundary}}.$$

$$\tag{16}$$

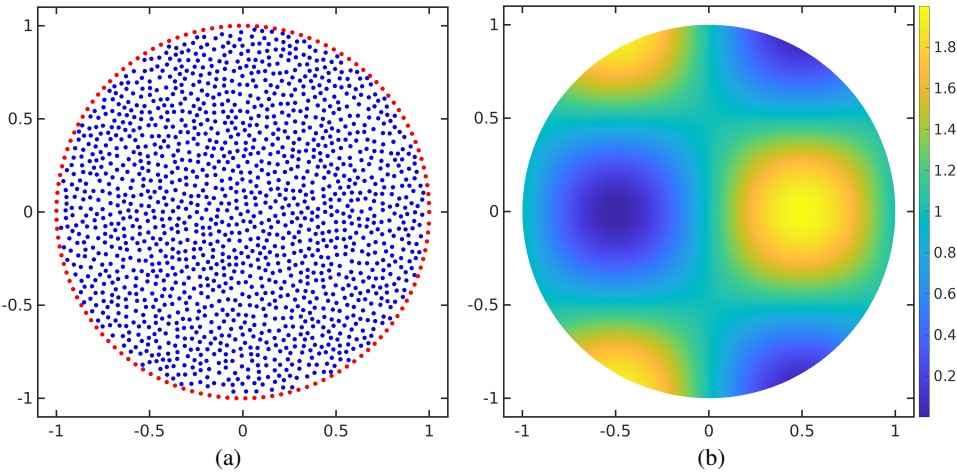

(a)    (b)

Figure 1: Quasi-uniform collocation points and the manufactured solution on the unit disk. The figure shows (a) $N = 1663$ interior and boundary collocation points on the unit disk and (b) the manufactured solution given by (19).

## 4 Results

We now present experimental results comparing DT-PINN and vanilla-PINN performance on the linear Poisson equation (3), the nonlinear Poisson equation (1), and the forced heat equation (5).

**Setup** All experiments were run for 5000 epochs on an NVIDIA GeForce RTX 2070. All results are reproducible with the seeds we used in the experiments. We used the *L-BFGS* optimizer with manually fine-tuned learning rates for both vanilla-PINNs and DT-PINNs. Both DT-PINNs and vanilla-PINNs used a constant NN depth of $s = 4$ layers with 50 nodes each across all runs. We use quasi-uniform collocation points generated using a node generator (30). Figure 1b shows the manufactured solution as specified in (19). For the Poisson experiments, we report errors on a test set of $N_{test} = 21748$ points. For the heat equation, we report results directly at the collocation points for convenience. For all experiments, the spatial domain $\Omega$ is set to the unit disk

$$\Omega = \{x \in \mathbb{R}^d \mid \|x\|_2^2 \le 1\}. \tag{17}$$

For illustration, we show one of the point sets on the unit disk in Figure 1a (with $N = 1663$ points). In the 2D heat equation experiment, the space-time domain is chosen to be $\Omega \times [0, 1]$. The time interval $[0, 1]$ is evenly divided into 24 time steps so that $N_t = 24$ (excluding $t = 0$), and the time-step was set to $\triangle t = \frac{1}{24}$. We measure all errors against a *manufactured* (specified) solution $u$, and specify $f$ so that the solution holds true. The boundary condition term $g$ is computed by applying the operator $\mathcal{B}$ to $u$; we use $\alpha = \beta = 1$ for all tests. To compare DT-PINNs and vanilla-PINNs to the manufactured solution $u$, we report the relative $\ell_2$ error

$$e_{\ell_2} = \frac{\|\tilde{\boldsymbol{u}} - \boldsymbol{u}\|_2}{\|\boldsymbol{u}\|_2}, \tag{18}$$

where $\boldsymbol{u}$ is the true solution vector, and $\tilde{\boldsymbol{u}}$ is either the DT-PINN or vanilla-PINN solution vector.

### 4.1 Effect of neural network depth

We first study the effect of PINN depth (fixing the number of nodes per layer) $s$ on computing the Laplacian $\Delta$ of the output with respect to the spatial variable $x$ using either autograd or RBF-FD. We compute errors against fp64 autograd for fp32 autograd and for RBF-FD with $p = 2, 3, 4$, and $5$. All errors were computed on $N = 19638$ quasi-uniform collocation points. The results are in shown in Figure 2a. We see $p = 4$ and $p = 5$ are more accurate than fp32 autograd, and that increasing $p$ increases the accuracy of RBF-FD by about two orders of magnitude. The errors are reasonably low for $p = 3$ also. In Figure 2b, we report the time taken for the same test. It is immediately clear that fp64 autograd is significantly more expensive than the fp32 variant, though both costs scale slowly with the network depth $s$. More importantly, the time taken for fp64 RBF-FD (for all orders) is both lower than both fp32 and fp64 autograd and is independent of the network depth $s$, primarily since the RBF-FD weights can be precomputed and repeatedly reused during training.

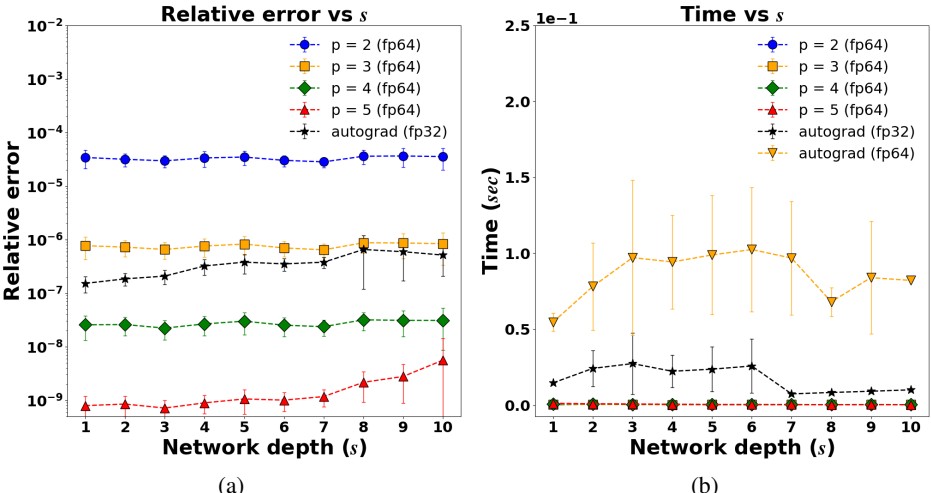

(a)         (b)

Figure 2: Autograd properties as a function of network depth $s$. The figure shows (a) effect of neural network depth $s$ on the relative error (with respect to fp64 autograd) and (b) time taken for one application of autograd on fp32 and fp64, compared to the time taken for SpMV using RBF-FD. The RBF-FD weights for $N = 19638$ collocation points were precomputed using an efficient CPU code in approximately $0.1s$. Error bars over 15 random runs are shown.

## 4.2 Linear Poisson equation

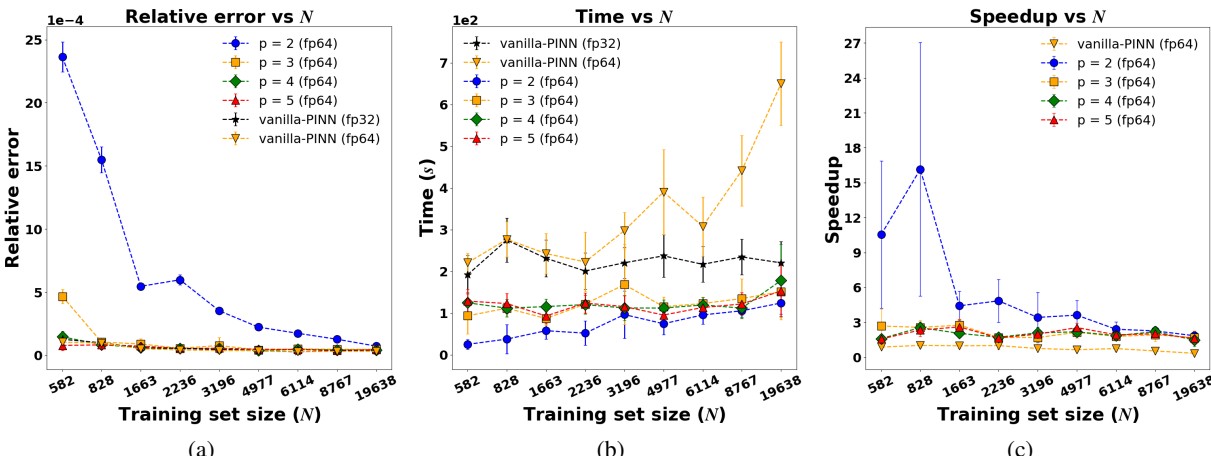

(a)         (b)         (c)

Figure 3: fp64 DT-PINNs and fp32 vanilla-PINN results on the linear Poisson equation (3) for different numbers of collocation points ($N$) and orders of accuracy ($p$). We show (a) the relative error in the PINN solution; (b) the time taken to converge to lowest relative error; and (c) the speedup attained by fp64 DT-PINNs relative to fp32 vanilla-PINN for those times. Error bars over 5 random runs are shown.

Next, we study the performance of fp64 DT-PINNs and fp32 vanilla-PINNs on the linear Poisson equation (3) on the domain (17). Letting $x = [x_1, \ x_2]$, we specify the true solution $u$ to be

$$u(x) = u(x_1, x_2) = 1 + \sin(\pi x_1)\cos(\pi x_2), \tag{19}$$

and enforce this by setting $f = \Delta u$; the true solution $u$ is shown in Figure 1b. We then solve for $\tilde{u}$ as described in Section 3. The results of this experiment are shown in Figure 3. We present relative errors (Figure 3a), wall clock time (Figure 3b), and speedup (Figure 3c). We also present results for fp64 vanilla PINNs. It is important to note that fp64 DT-PINNs were completely stored and trained in fp64, a format widely known to be significantly slower on the GPU than fp32.

Figure 3a shows the relative errors for DT-PINNs as a function of the number of collocation points $N$. DT-PINNs for $p = 3, 4, 5$ produce similar relative errors to both fp32 and fp64 vanilla-PINNs for the same value of $N$. In contrast, the DT-PINN using $p = 2$ is generally less accurate, showing that higher-order accuracy is needed to reach the same relative errors as vanilla-PINNs. Examining Figures 3b and 3c, we also see that all fp64 DT-PINNs can be trained much more rapidly than both fp32 and fp64 vanilla-PINNs. In fact, Figure 3c shows a **maximum training speedup of 4x for DT-PINNs even if $p = 2$ is ignored**. **In general, fp64 DT-PINNs for $p > 2$ are trained much more quickly than vanilla-PINNs without a significant loss in accuracy**. We also note that using fp32 DT-PINNs did not lead to greater speedups over the fp64 DT-PINNs, with a loss in accuracy. These results are shown in Appendix A.1.2.

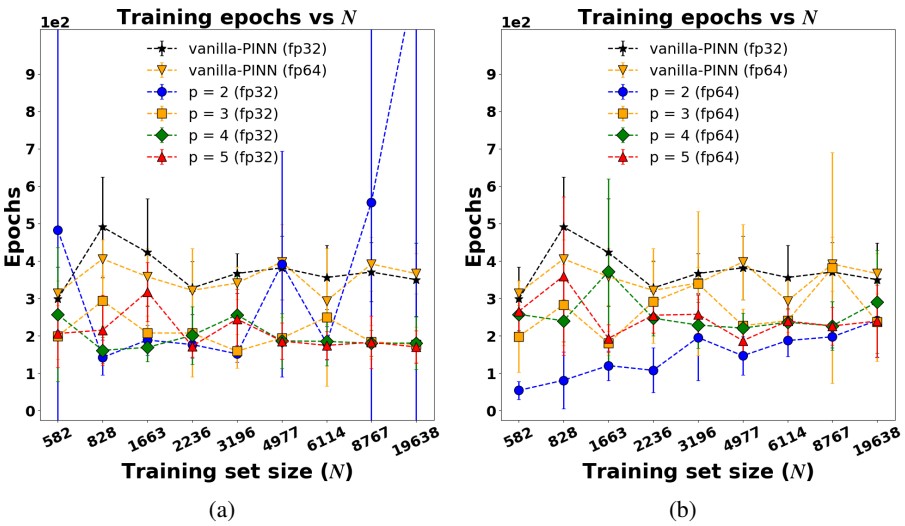

(a)  (b)

Figure 4: Number of training epochs to achieve the lowest relative error as a function of number of collocation points $N$ and order $p$ for (a) fp32 DT-PINNs and (b) fp64 DT-PINNs. Error bars over 5 random runs are shown.

The superior performance of fp64 DT-PINNs becomes clearer when we examine the number of training epochs as a function of the number of collocation points $N$ (Figure 4). Figures 4a and 4b both illustrate that both fp32 and fp64 DT-PINNs reach their lowest relative errors in fewer epochs than vanilla-PINNs. These results provide evidence that DT-PINNs have simpler loss function landscapes than their vanilla-PINN counterparts, also implying that loss functions involving linear combinations of NN values are easier to minimize than loss functions involving derivatives of NNs. Figure 4b also shows that only fp64 DT-PINNs take fewer epochs to train as $N$ is increased. We also see that moving to fp64 does not appear to significantly speed up vanilla-PINNs. It is therefore the *combination* of discrete training and fp64 that results in speedups for increasing $N$.[1]

### 4.3 Nonlinear Poisson equation

Next, to understand the influence of nonlinearities in terms not including the differential operator, we test the performance of DT-PINNs on the nonlinear Poisson equation (1). To measure errors, we use the manufactured solution given by (19), and set $f = \Delta u - e^u$. The results are shown in Figure 5; for simplicity, we omit $p = 2$ and fp64 vanilla-PINNs as both these have poor cost-accuracy tradeoffs. First, Figure 5a shows that despite some outliers, fp64 DT-PINNs achieve comparable relative errors to fp32 vanilla-PINNs. Further, Figure 5b shows that DT-PINNs are still trained faster than vanilla-PINNs. However, when comparing Figure 5c to Figure 3c (linear Poisson equation), we see that the average speedup is higher for the linear Poisson equation. **This shows that DT-PINNs may not offer speedups over vanilla-PINNs if terms not involving differential operators dominate training times**.

---

[1]We also attempted to train fp32 vanilla-PINNs using ghost points, but using ghost points offered no improvement (results not shown).

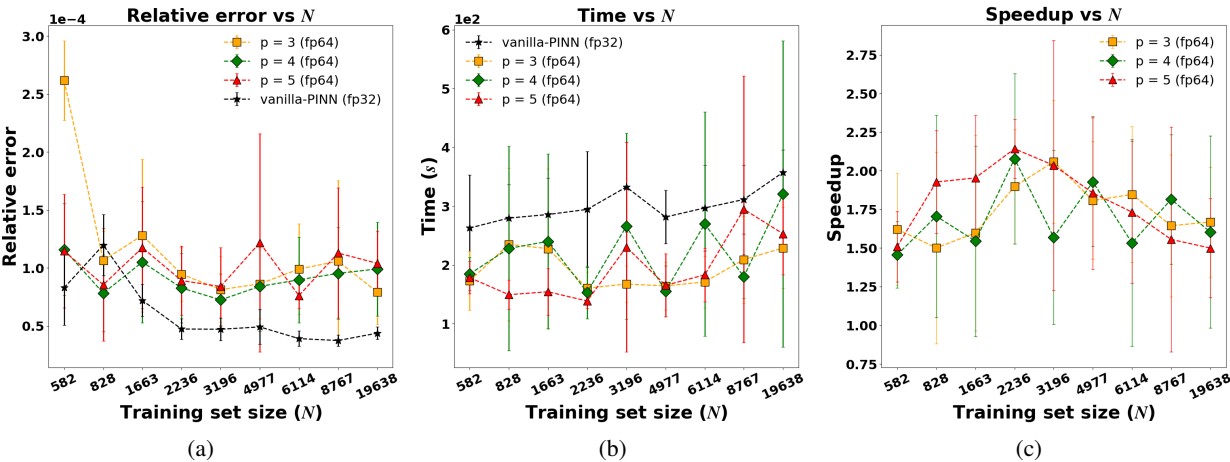

(a)                    (b)                    (c)

Figure 5: fp64 DT-PINNs and fp32 vanilla-PINN results on the nonlinear Poisson equation (1) for different numbers of collocation points ($N$) and orders of accuracy ($p$). We show (a) the relative error in the PINN solution; (b) the time taken to converge to lowest relative error; and (c) the speedup attained by fp64 DT-PINNs relative to fp32 vanilla-PINN for those times. Error bars over 5 random runs are shown.

## 4.4 Heat equation

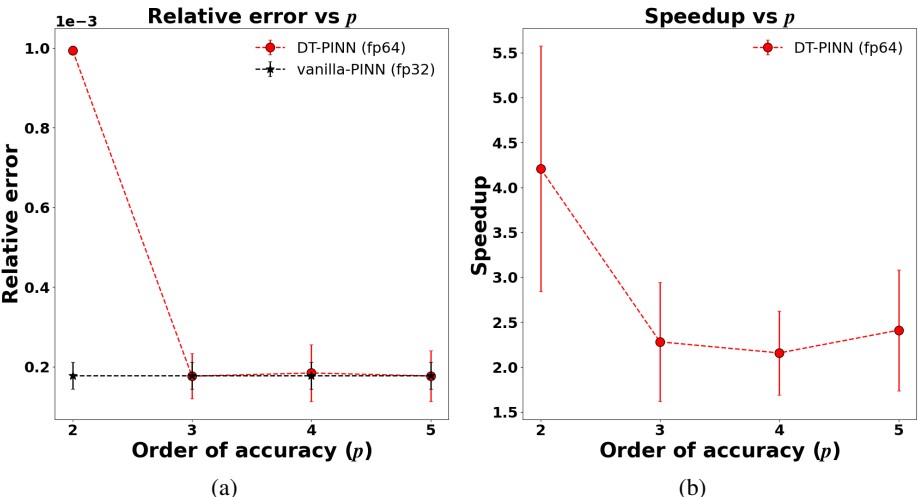

(a)                          (b)

Figure 6: fp64 DT-PINN and fp32 vanilla-PINN results on the heat equation on $N = 828$ spatial points and $N_t = 24$ time-steps. The figure shows (a) the relative error for fp64 DT-PINNs as a function of approximation order $p$; and (b) the speedup attained by fp64 DT-PINNs over fp32 vanilla-PINNs as a function of $p$. Error bars over 5 random runs are shown.

Next, we compare fp64 DT-PINNs and fp32 vanilla-PINNs on the 2D heat equation. In order to demonstrate the flexibility of our method, we adopt a mixed training approach where only spatial derivatives are discretized with RBF-FD. We specify the true solution $u$ to be

$$u(x, t) = u(x_1, x_2, t) = 1 + \sin(\pi x_1) \cos(\pi x_2) \sin(\pi t), \tag{20}$$

and specify $f = \frac{\partial u}{\partial t} - \Delta u$ so that the solution $u$ satisfies the heat equation for all space-time. We compute the initial condition as $u_0(x, 0) = u(x_1, x_2, 0) = 1$. We trained on $N = 828$ spatial collocation points over 25 time slices (including time $t = 0$) for a total of $20,700$ spacetime collocation points; we express all results as a function of $p$. These results are shown in Figure 6.

First, Figure 6a shows similar results to the 2D Poisson equation, with $p > 2$ achieving relative errors similar to fp32 DT-PINNs. Figure 6b shows that we achieve 2-4x speedups over vanilla-PINNs. We observed in our experiments that the speedup appears to increase as a function of the number of time-steps $N_t$ (results not shown). It is likely that one could achieve further speedups by also discretizing the temporal derivatives, but we leave this exploration for future work.

## 5 Summary and future work

We presented a novel technique, DT-PINNs, that involves training PINNs by using RBF-FD for spatial derivatives, and using fp64 weights and training instead of fp32. This involved replacing all autograd operations (dense matrix-matrix multiplies) related to PDE loss terms with an SpMV operation. We showed that using an RBF-FD approximation order of $p > 2$ resulted in DT-PINNs that were comparable in accuracy to vanilla-PINNs while offering 2-4x speedups in training times for both the linear and nonlinear Poisson equations. We also showed that DT-PINNs trained in a mixed fashion (autograd for time, RBF-FD for space) also achieved comparable accuracy and speedup on the heat equation. DT-PINNs therefore constitute a new paradigm for scientific machine learning that allow practitioners to leverage existing sophisticated scientific computing techniques to accelerate ML training times.

There are several possible extensions to our current work. It is likely that using DT-PINNs in conjunction with X-PINNs and G-PINNs will yield even greater speedups in training times. Further, DT-PINNs open the door to leveraging compute more efficiently. For instance, the SpMV operations could be parallelized using distributed memory systems in conjunction with GPUs, thereby allowing scaling to very large training sets; alternatively, the SpMV operation could be parallelized on many-core CPUs while other operations are conducted on the GPU. It may also be profitable to explore mixed-precision training of DT-PINNs. Finally, DT-PINNs can be viewed as vanilla-PINNs with partially linearized constraints; it may be profitable to explore other types of constraint linearization to accelerate training and simplify loss function landscapes.

## Acknowledgments and Disclosure of Funding

VS was supported by National Science Foundation (NSF) grant CCF 1714844.

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
