# OpenReview forum: "Accelerated Training of Physics-Informed Neural Networks (PINNs) using Meshless Discretizations"
_NeurIPS.cc/2022/Conference — NeurIPS 2022 Accept_

### Official Review · Reviewer_g9sM · 2022-06-28

**Rating:** 7
**Confidence:** 3
**Soundness:** 3 good
**Presentation:** 3 good
**Contribution:** 3 good

**Summary:**


The submitted work proposes the use of a discrete finite difference (FD) method for a faster computation of spatial derivatives (i.e. with respect to the signal domain) in Physics Informed Neural Networks (PINNs).
More specifically, the use of Radial-Basis-Function FD (RBF-FD) is proposed, which approximates an operator at a given position $x_i$ via a weighted linear combination of a position $x_i$ and its neighbors.
The weights can be learned by solving a dense block linear system, and an arbitrary point cloud may be used.
The underlying (implicit) interpolation between the points is a combination of RBFs and Legendre polynomials.
The linear operators are precomputed so that during the PINN training they can be directly evaluated on the collocation points.
Experiments show the application of RBF-FD for the linear and non-linear Poisson equation, as well as the two-dimensional heat equation. For the heat equation, the temporal derivatives are still computed using auto-grad, and only the spatial derivatives are approximated via RBF-FD.

Results show that the proposed method is able to accurately approximate the derivatives of Neural Networks with a significant speed-up compared to automatic differentiation. The speed-up depends on the complexity of the approximated operator.
Applied to PINNs, this results in faster training of the networks with comparable accuracy to vanilla PINNs. Furthermore, the use of RBF-FD required fewer training epochs than vanilla PINNs, however, the exact causes for this are not known.

Contributions:
- The use of RBF-FD for approximating differential operators in PINNs is proposed as an alternative to autograd
- It is empirically shown that, with a sufficiently high order, RBF-FD provides accurate and fast approximations to the Laplacian of a Neural Network
- On experiments for the (non-)linear Poisson equation and the Heat equation a  significant speed-up of PINNs is demonstrated. Fewer epochs were required for reaching a similar error as vanilla PINNs.

**Questions:**

- Going back to my point in the "Quality" section, why are only spatial derivatives approximated with RBF-FD?

- Is there any existing understanding or intuition regarding **why** the RBF-FD method requires fewer training epochs?

**Limitations:**

- The authors highlight that the speedup depends on the complexity of the approximated operator, i.e. they may not offer much speedup if nonlinear terms not involving differential operators dominate training times.
- The approximations to the operators have to be precomputed.

**Strengths And Weaknesses:**

### Significance:
- [High] The authors provide an efficient alternative to autograd for computing PDE losses in PINNs.
- [Low] The proposed method results in fewer required epochs for attaining a similar loss compared to vanilla PINNs.
- [Low] The proposed method appears to be very general and orthogonal to existing methods for speeding up PINNs, allowing a seamless combination with existing work.

### Originality:
- The usage of RBF-FD as an alternative to autograd for PINNs is to the best of my knowledge novel.
- As far as I can judge, RBF-FD methods seem to be well-established within the numerical PDE literature but have not yet been explored much within Machine Learning.

### Quality:
- The submission and experimental settings appear to be sound to the best of my knowledge.
- While limited in their extent (in terms of different PDEs), the experiments serve as a valuable proof of concept.
- It is not explained, why the approximation is only used for spatial derivatives whereas autograd is used for temporal derivatives.
It is unclear, whether this serves as an example for mixed usage of autograd and RBF-FD, or is due to some limitation of the underlying method.
- Well maintained and readable code is provided; The appendix provides implementation details for the autograd implementation of sparse matrix operations, that are required for the proposed method.
- No error bars are provided (Checklist 3.c). The authors argue that this is the norm for PINNs, which in my opinion does not warrant ignoring general best practices within the ML community. However, another argument made is the limited computational resources, which is reasonable for the PINN experiments. Experiments concerning Figure 1 (autograd vs RBF-FD) should not be very compute-intensive.


### Clarity:
- The paper is generally well written and argued for.
- (minor) The plots are at times a bit confusing:
    - The x and y labels could be a bit bigger and more expressive (e.g. "network depth s" instead of just "s" in Figure 1, similar to N in Figures 2-4 and p in Figure 5)
    - It's hard to (quickly) distinguish the vanilla PINN from the proposed method. Maybe a different style or color coding would help.


To summarize,  the paper provides an interesting and feasible alternative for computing spatial derivatives in PINNs, bridging the gap between well-established mesh-free methods for PDEs and PINNs.
The proposed method is novel and to the best of my knowledge technically sound, experiments are reasonable and reproducible, and the paper is well written.

---

> ### Author Response · Authors · 2022-08-02
> **Addressing comments under "Quality"**
>
> We thank the reviewer for their comments. All revisions addressing this reviewer's comments will be colored blue, except where they overlap with a different reviewer (those are colored red or cyan).
>
> We now address the following comments:
>
> 1. "It is not explained, why the approximation is only used for spatial derivatives whereas autograd is used for temporal derivatives. It is unclear, whether this serves as an example for mixed usage of autograd and RBF-FD, or is due to some limitation of the underlying method."
>
> We apologize for this oversight. We chose this mixed training technique to serve as an example of the flexibility of our method, not due to any underlying limitations. However, coincidentally, using mixed training allowed us to bypass CFL restrictions while training (though this was not our original intent for choosing mixed training). The text in Section 3 under "heat equation" has been edited to clarify these points. We have also added sentences to Section 4.4 clarifying this.
>
> 2. "No error bars are provided (Checklist 3.c). The authors argue that this is the norm for PINNs, which in my opinion does not warrant ignoring general best practices within the ML community. However, another argument made is the limited computational resources, which is reasonable for the PINN experiments. Experiments concerning Figure 1 (autograd vs RBF-FD) should not be very compute-intensive."
>
> We agree with the reviewer, and now provide error bars over 15 runs for the autograd test (Figure 1), and 5 runs for all the other simulations. This was the best we were able to do due to time and resource constraints. We have also modified the Checklist to indicate this change.

---

> > ### Comment · Reviewer_g9sM · 2022-08-08
> > **Response**
> >
> > Thank you for the detailed response, which resolves my concerns regarding the experimental settings - especially the text changes in the revised version.

---

> > > ### Author Response · Authors · 2022-08-09
> > > **Response to "Response"**
> > >
> > > Thank you for your helpful suggestions. We have now also added error bars for the fp32 results in the appendix.

---

> ### Author Response · Authors · 2022-08-02
> **Addressing "Questions"**
>
> We address the following reviewer comments under the "Questions" section:
>
> 1. "Going back to my point in the "Quality" section, why are only spatial derivatives approximated with RBF-FD? "
>
> See our comment addressing this under "Quality". We only did this to illustrate that mixed training was possible. In future work, we plan to explore full spatio-temporal discretizations.
>
> 2. "Is there any existing understanding or intuition regarding why the RBF-FD method requires fewer training epochs?"
>
> A different reviewer also pointed out that this was missing from the paper, and the comment addressing this has been added in the color red to Section 4.2. To the best of our knowledge, the literature does not provide an answer to this. However, we believe that DT-PINNs require fewer epochs primarily because the use of RBF-FD results in a loss function that is a straightforward linear combination of the DT-PINN values at spatial locations. The gradient of this loss with respect to the DT-PINN weights is likely easier to minimize that the gradient of the vanilla-PINN loss, which contains (analytical) derivatives of the vanilla-PINNs. The latter are likely complicated nonlinear functions, while RBF-FD can be thought of as a partial linearization of this loss function. We make a similar comment at the end of our conclusion section (last sentence of the paper).

---

> ### Author Response · Authors · 2022-08-02
> **Addressing "Clarity"**
>
> "(minor) The plots are at times a bit confusing:
>
>     The x and y labels could be a bit bigger and more expressive (e.g. "network depth s" instead of just "s" in Figure 1, similar to N in Figures 2-4 and p in Figure 5)
>     It's hard to (quickly) distinguish the vanilla PINN from the proposed method. Maybe a different style or color coding would help."
>
> We have made the axis labels more expressive. We have also tried to distinguish vanilla PINNs by using a different color.

---

### Official Review · Reviewer_8h43 · 2022-07-11

**Rating:** 8
**Confidence:** 4
**Soundness:** 4 excellent
**Presentation:** 3 good
**Contribution:** 4 excellent

**Summary:**

The paper uses a linear combination of radial bias functions  (RBFs) with additive odd poly-harmonics to discretise partial differential equations in a meshless manner. A deep neural network is used as trial function to obtain the approximate solutions through training over the data points with the equation to be solved as soft constraints.  In the  RBF-FD (finite differences) method the calculation of derivatives is performed with precomputed stencils. Restricting the approximation to nearest neighbors reduces the computational load significantly. The paper displays a solutions of Poisson and Heat dissipation equations as examples of the strengths of the method. It is found that the using a 4-ayer deep neural network that is trained to fulfill the equations using  GPU fp64 is very efficient in solving the problem compared to the state of the art.

**Questions:**

Question:
In the heat equation, the spatial data points are not in a regular mesh, but the mesh in time dimensions is. In the approximation (8) of the difference matrix in the implementation of the RBF-DT, the spatial stencil is based on nearest neighbors. How one is ensuring that the local Courant number is in the correct range (i.e CFL conditions are always fulfilled) despite the varying size of the spatial stencil? Is the global time step chosen in relation to the smallest spatial neighborhood in the irregular mesh? Or is the number of nodes in the spatial stencil made dependent of its local physical size? To me it seems that the RBF-DT with the full difference matrix is an implicit solver where this problem is not necessarily emerging, but making the approximation (8) makes the solver explicit as it will not depend on data further away of the nearest neighborhood.  Or is the fact that the temporal derivatives are calculated analytically with autograd using equation (15) automatically solving the issue?
Could this be clarified in the text for the final version?

Suggestion:
The reference to equations (3) and (2) on line 113 should be (2) and (1).

**Limitations:**

I do not think there are any potential negative societal impacts of this work.

**Strengths And Weaknesses:**

The paper goes trough, in a clear way, the theory of the  physics informed neural networks (PINN= and then builds upon that to introduce RBF-DF in a way that is pretty easy to follow.  It described system addresses many of the problems that arise in  the work of solving PDEs for complicated (and practical) geometries. The meshless nature  of the method makes its able to handle complex cases. Also, as it lacks the usual meshing of the geometry it is  much faster to set up ready for the computation, in addition to being efficient in its calculations.

Of course, one would like that PyTorch would have all the  required autograd functions already implemented for even simpler scripting, but this is no way fault in the paper, but comment on paradigm shift where AI is providing new capabilities and better and better tools for problem solving. The appendix and submitted materials are valuable and greatly appreciated by those planning to use the solution method.

The figures do not have error bars, so one thinks the performance gains are reported from a single training run, albeit there is a good set of comparisons done to state of the art.  I would like to see results gathered from multiple runs for each of the points.

---

> ### Author Response · Authors · 2022-08-02
> **Addressing "Weaknesses"**
>
> We thank the reviewer for their comments. All revisions to the paper for this reviewer are colored cyan.
>
> We address the following reviewer comment:
>
> "The figures do not have error bars, so one thinks the performance gains are reported from a single training run, albeit there is a good set of comparisons done to state of the art. I would like to see results gathered from multiple runs for each of the points."
>
> We now show results gathered from multiple runs for each of the points in the paper's main results section. Due to time and resource constraints, we were only able to use 5 runs for the PDE simulations, but were able to use 15 runs for the autograd test in Figure 1. We also believe that averaging out over more runs will further shrink the error bars as they were already quite small (see Figure 2a for instance).  We were not able to add error bars to the new results in the appendix, once again due to time and resource constraints.

---

> > ### Author Response · Authors · 2022-08-09
> > **Update**
> >
> > We have now added error bars to the results in the appendix also.

---

> ### Author Response · Authors · 2022-08-02
> **Addressing "Questions"**
>
> We address the following question from the reviewer:
>
> "Question: In the heat equation, the spatial data points are not in a regular mesh, but the mesh in time dimensions is. In the approximation (8) of the difference matrix in the implementation of the RBF-DT, the spatial stencil is based on nearest neighbors. How one is ensuring that the local Courant number is in the correct range (i.e CFL conditions are always fulfilled) despite the varying size of the spatial stencil? Is the global time step chosen in relation to the smallest spatial neighborhood in the irregular mesh? Or is the number of nodes in the spatial stencil made dependent of its local physical size? To me it seems that the RBF-DT with the full difference matrix is an implicit solver where this problem is not necessarily emerging, but making the approximation (8) makes the solver explicit as it will not depend on data further away of the nearest neighborhood. Or is the fact that the temporal derivatives are calculated analytically with autograd using equation (15) automatically solving the issue? Could this be clarified in the text for the final version?"
>
> This is an excellent question! As the reviewer suspects, we believe that the use of autograd obviates the need for a CFL-type constraint on the time-step and node spacing. The CFL constraint arises as an interplay between the inherent stability region of time-integration schemes (such as Runge-Kutta and multistep methods) and the details of the spatial discretization. Since our "time integration" is exact, we believe no CFL constraint is needed. A comment clarifying this has been added to Section 3 (heat equation).
>
> However, to fully address the spirit of the reviewer's comment, we also remark that in other work involving RBF-FD where CFL constraints are important (see reference [29] for instance), it is common to use quasi-uniformly distributed collocation points in space, and evenly-spaced points in time, and then select the time-step according to the CFL constraint to ensure stability. In this setting, it appears to be common to use either average sample spacing or the smallest distance between any pair of samples.
>
> To the best of our knowledge, it is common in the PINN literature to use high-order implicit Runge-Kutta methods to bypass CFL-type constraints (see for example Section 3 in https://arxiv.org/abs/1711.10561). This should be easily doable in the context of DT-PINNs also.
>
> "Suggestion: The reference to equations (3) and (2) on line 113 should be (2) and (1)."
>
> We thank the reviewer for catching this, and have made the change.

---

### Official Review · Reviewer_6bVr · 2022-07-11

**Rating:** 4
**Confidence:** 4
**Soundness:** 2 fair
**Presentation:** 3 good
**Contribution:** 2 fair

**Summary:**

The paper leverages modern scientific computing techniques, specifically the meshless radial basis function-finite difference (RBF-FD) method, to accelerate the training of PINNs. In detail, the paper replaces the computationally expensive automatic differentiation by RBF-FD with sparse-matrix vector multiplication, allowing a high efficiency and accuracy computation of the partial derivative terms, and a fp64 training on GPU. Several numerical experiments are carried out, including linear, non-linear and space-time problems, showing that the proposed method is significantly faster than fp32 vanilla-PINNs with comparable accuracy.

**Questions:**

1) The paper claims in line 77-78 that the proposed method “can be trained without any loss in accuracy on irregular domains (just as autograd does)”, while all the evaluation cases are carried out on regular domains, i.e. a unit disk in 2D. Extra evaluation cases with irregular domains are recommended.

2) It is agreed that “the repeated computation of the partial derivative terms in the PINN loss functions via automatic differentiation during training is known to be computationally expensive, especially for higher-order derivatives”. Is the RBF-FD method capable to handle higher-order derivatives? If yes, it is recommended to add extra evaluation cases with higher order PDEs.

3) In section 2.2 the RBF method is explained, while it is not quite clear the computation power required for pre-computation of the RBF-FD weights. For example, in Figure1(b), the time for pre-computing the RBF-FD weights should also be considered to achieve a fair comparison between autograd and SpMV with RBF-FD.

4) Considering the RBF-FD weights, it is recommended to declare whether it is required to recompute and update during the training, especially for complex problems, e.g. strongly non-linear problem, shape deformation/optimization problem?

**Limitations:**

This method offers limited speed-up if the nonlinear terms not invoking differential operations dominates the calculation time.

**Strengths And Weaknesses:**

Strengths:
The idea of the paper is straight forward - replacing the automatic differentiation by RBF-FD and sparse matrix-vector multiplication. The theoretical grounding is explained clearly, several empirical evaluations are used to validate the proposed method. The novelty of the contribution is fair, although both the PINNs and RBF-FD are well developed techniques, this is the first work combining them together to accelerate the PINNs training. The topic is closely related to the NeurIPS community.

Weaknesses:
The weakness of the paper is mainly that the evaluation cases considered are somehow naive, which could not demonstrate the advantages of the proposed method sufficiently. The paper barely talks about the motivation of choosing fp64 over fp32 in the first place, and gives little reasons and evidences about why it would cause a training speed-up. And this leads to my suspicion on whether choosing fp64 would become a key point.

---

> ### Author Response · Authors · 2022-08-02
> **Addressing "Weaknesses"**
>
> We thank the reviewer for their comments. All revisions to the paper for this reviewer are colored red.
>
> We address the following comment:
> "The weakness of the paper is mainly that the evaluation cases considered are somehow naive, which could not demonstrate the advantages of the proposed method sufficiently. The paper barely talks about the motivation of choosing fp64 over fp32 in the first place, and gives little reasons and evidences about why it would cause a training speed-up. And this leads to my suspicion on whether choosing fp64 would become a key point."
>
> We apologize if the paper was insufficiently clear. Our motivation for choosing fp64 over fp32 is to provide a speedup over fp32 vanilla-PINNs without a degradation in accuracy. Our revisions hopefully make this a bit clearer, but the line of reasoning is as follows:
> 1. Figure 1 shows that SpMV with a precomputed RBF-FD matrix is cheaper than autograd.
>
> 2. Figure 2 now shows that fp64 DT-PINNs obtain the same accuracy as both fp32 and fp64 vanilla-PINNs, but that the latter two are significantly slower. The speedup appears to not be just from moving to fp64, but also because we use discrete training.
>
> 3. Figure 3 attempts to tease out the source of the speedup by measuring the number of training epochs. Using fp64 for vanilla-PINNs results in a small reduction in the number of epochs over fp32 vanilla-PINNs, but not much. In contrast, both fp32 and fp64 DT-PINNs achieve the lowest error in fewer epochs, showing that there is indeed a benefit to using discrete training.
>
> 4. While the reason for this reduction in the number of epochs is not completely clear to us, we provide a possible explanation for this in the text in Section 4.2. Essentially, use DT-PINNs use loss functions that are a linear combination of NN values, while vanilla-PINNs use loss functions that are (effectively) the analytical derivatives of the NN. It appears that the latter is more challenging to optimize than the former. We hope to investigate this in future work.
>
> 5. Figure 7 added to the appendix shows that fp32 DT-PINNs do obtain a speedup over both fp32 and fp64 vanilla-PINNs, but with a degradation in accuracy. This also adds to the evidence that it is not merely the move to fp64 that results in greater speedup, but the combination of fp64 and RBF-FD. fp64 DT-PINNs are in our opinion the superior choice (when compared to their fp32 counterparts).

---

> ### Author Response · Authors · 2022-08-02
> **Addressing "Questions"**
>
> This comment addresses Questions 1-4.
>
> 1. "The paper claims in line 77-78 that the proposed method “can be trained without any loss in accuracy on irregular domains (just as autograd does)”, while all the evaluation cases are carried out on regular domains, i.e. a unit disk in 2D. Extra evaluation cases with irregular domains are recommended"
>
> We apologize for the lack of clarity. In our experience, in the numerical PDE literature, the phrase "irregular domains" is often synonymous with the phrase "domains with curved boundaries". For instance, traditional finite difference methods have difficulties in handling such boundaries, while RBF-FD, finite elements, and PINNs do not. We thus wished to select the most intuitive test case that illustrated the ability of our method to handle curved boundaries, and therefore chose the disk. In the absence of space and time constraints, we would be happy to supply more test cases on other domains. However, we do note that the RBF-FD literature has many examples of RBF-FD being used without any particular challenge on more irregular domains than the disk. See for instance reference [30] in our article. For transparency, we have edited the text to replace the phrase "irregular domains" with "domains with curved boundaries".
>
> 2. "It is agreed that “the repeated computation of the partial derivative terms in the PINN loss functions via automatic differentiation during training is known to be computationally expensive, especially for higher-order derivatives”. Is the RBF-FD method capable to handle higher-order derivatives? If yes, it is recommended to add extra evaluation cases with higher order PDEs."
>
> The RBF-FD method can indeed handle higher-order derivatives. While we did not solve a higher-order PDE due to time constraints, we now present a version of the autograd test for computing the biharmonic operator (Laplacian squared) in the Appendix (Figure 6). The speedups over autograd are even more obvious in this case, since this operator contains both fourth and second order spatial derivatives.
>
> 3. "In section 2.2 the RBF method is explained, while it is not quite clear the computation power required for pre-computation of the RBF-FD weights. For example, in Figure1(b), the time for pre-computing the RBF-FD weights should also be considered to achieve a fair comparison between autograd and SpMV with RBF-FD"
>
> We respectfully disagree that the RBF-FD precomputation time should be added to Figure 1(b). Since PINN training can take hundreds or even thousands of epochs to achieve a satisfactory tolerance, we believe that the time to precompute RBF-FD weights is more than dominated by the cost of spatial and temporal autograd operations during a typical training loop (on a stationary domain).
>
> However, in the interest of fairness, we have added a note to the caption of Figure 1 indicating the time for precomputation of RBF-FD weights using a CPU (though this computation can be easily parallelized on the GPU, we do not currently have a GPU parallelized code). This cost can then be taken into account by readers/practitioners when solving problems that require recomputation of RBF-FD weights.
>
> 4. "Considering the RBF-FD weights, it is recommended to declare whether it is required to recompute and update during the training, especially for complex problems, e.g. strongly non-linear problem, shape deformation/optimization problem?"
>
> Excellent point. We have added a comment to the RBF-FD section clarifying this. Recomputation is indeed needed on problems involving moving domains/shape deformation, though one can get away with recomputing weights purely in the neighborhood of the deformation (see reference [32]).

---

> ### Author Response · Authors · 2022-08-09
> **Irregular domains**
>
> We now have some limited results on an irregular domain in the appendix.

---

### Author Response · Authors · 2022-08-09
**Summary of revisions**

Here, we summarize the revisions made in response to reviewer comments.

1. All our plots now have error bars.
2. A more detailed justification for the speedups has been added to Section 4.2, discussing in greater detail how fp64 + discrete training is the reason for the superior cost-accuracy profile of DT-PINNs to vanilla-PINNs (both fp32 and fp64). While fp32 + discrete training also offers speedups, it suffers from a degradation in accuracy compared to vanilla-PINNs (both fp32 and fp64).
3. The text has been edited to clarify details about time-step size and CFL constraints.
4. A comparison of RBF-FD to autograd (analogous to Section 4.1) for the Biharmonic operator (a fourth order differential operator) has been added to the appendix.
5. fp32 DT-PINN results (and comparisons to fp32 and fp64 vanilla PINNs) have been added to the appendix.
6. Clarification of the computation time of RBF-FD weights has been added (Figure 1 caption), as have comments on when recomputation might be necessary (Section 2.2).
7. As reviewer 6bVr requested results on a more irregular domain than the disk, we have some results for DT-PINNs on a star-shaped domain in the appendix for small values of N. We still obtain speedups, though we would obtain even larger ones for larger values of N.

We still note that the literature contains many examples of RBF-FD being used in irregular domains without any modification to the formulation presented in the paper.

---

### Meta-Review · Area_Chair_X7o9 · 2022-08-26

**Recommendation:** Accept
**Confidence:** Certain

**Metareview:**


All reviewers agreed that this paper has several strengths, such as a convincing motivation, a well structured and well-formulated model and solid theoretical grounding.
While two reviewers had a very positive general impression of the paper (emphasizing, in particular, the novelty and originality of this work), one reviewer raised some concerns about the application cases being too simplistic and not well suited for demonstrating potential strengths or weaknesses of the method. In my opinion, however, theses concerns (and further questions) could be addressed reasonably well in the rebuttal, and therefore, I recommend to accept this paper.



**Award:**

No

---

### Decision · Program_Chairs · 2022-09-14

Accept